# NAC Transcription Factor *GmNAC035* Exerts a Positive Regulatory Role in Enhancing Salt Stress Tolerance in Plants

**DOI:** 10.3390/plants14091391

**Published:** 2025-05-05

**Authors:** Wanting Shi, Sixin Ye, Yiting Xin, Hongmiao Jin, Meiling Hu, Yueping Zheng, Yihua Zhan, Hongbo Liu, Yi Gan, Zhifu Zheng, Tian Pan

**Affiliations:** The Key Laboratory for Quality Improvement of Agricultural Products of Zhejiang Province, College of Advanced Agricultural Sciences, Zhejiang A&F University, Hangzhou 311300, China; 2022101012015@stu.zafu.edu.cn (W.S.); 2022601022033@stu.zafu.edu.cn (S.Y.); xyt2023601022035@stu.zafu.edu.cn (Y.X.); jhm@stu.zafu.edu.cn (H.J.); hml@stu.zafu.edu.cn (M.H.); zhengyp@zafu.edu.cn (Y.Z.); yhzhan@zafu.edu.cn (Y.Z.); hbliu@zafu.edu.cn (H.L.); zjuganyi@163.com (Y.G.)

**Keywords:** soybean, NAC, transcription factor, salt tolerance, stress-related genes

## Abstract

Soybean, a globally significant and versatile crop, serves as a vital source of both oil and protein. However, environmental factors such as soil salinization pose substantial challenges to its cultivation, adversely affecting both yield and quality. Enhancing the salt tolerance of soybeans can mitigate yield losses and promote the development of the soybean industry. Members of the plant-specific transcription factor family NAC play crucial roles in plant adaptation to abiotic stress conditions. In this study, we screened the soybean *GmNAC* family genes potentially involved in the salt stress response and identified 18 *GmNAC* genes that may function during the early stages of salt stress. Among these, the *GmNAC035* gene exhibited a rapid increase in expression within one hour of salt treatment, with its expression being induced by abscisic acid (ABA) and methyl jasmonate (MeJA), suggesting its significant role in the soybean salt stress response. We further elucidated the role of *GmNAC035* in soybean salt tolerance. GmNAC035, a nuclear-localized transcriptional activator, enhances salt tolerance when overexpressed in Arabidopsis, reducing oxidative damage and boosting the expression of stress-responsive genes. It achieves this by regulating key stress response pathways, including the SOS pathway, calcium signaling, and ABA signaling. These findings highlight the potential of *GmNAC035* as a genetic engineering target to improve crop salt tolerance.

## 1. Introduction

Globally, soybean (*Glycine max* L.) ranks among the most significant crops. It plays a crucial role as an essential provider of oil and protein, which are used for human food and animal feed. Its economic and nutritional significance makes it a focal point of agricultural research, particularly in the context of enhancing yield and stress tolerance [1]. However, soybean production faces numerous challenges, among which abiotic stresses pose a significant threat to crop productivity. Salinity is one of the most prevalent environmental stresses. The intensification of soil salinization and the degradation of irrigated land directly affect dryland agriculture on 2.6 billion hectares worldwide. Projections suggest that by 2050, the primary and secondary salinization of agricultural soils will directly affect half of the world’s arable land, leading to substantial crop losses worldwide [2,3,4]. Therefore, understanding the molecular mechanisms of soybean’s response to salt stress is crucial for formulating strategies to improve its salt tolerance.

Salt stress has a negative impact on soybeans across various aspects, such as physiological, biochemical, and molecular procedures. High salinity disrupts ionic homeostasis, leading to the accumulation of ions such as sodium (Na^+^) and chloride (Cl^−^), which impair cellular functions and damage plant tissues. Additionally, salt stress induces oxidative stress, which then results in the overproduction of reactive oxygen species (ROS) that cause cellular damage. These physiological disruptions ultimately lead to reduced photosynthesis, stunted growth and development, and decreased yield [5,6,7]. Given the increasing salinization of arable land due to irrigation practices and climate change, there is an urgent need to identify and characterize genes that confer salt tolerance in soybeans in order to improve the salt tolerance and total yield of soybeans through breeding initiatives [8]. Among the various gene families involved in plant stress responses, the NAC (NAM, ATAF, and CUC) family has emerged as a key regulator of abiotic stress tolerance [9]. NAC proteins are marked by a well-conserved N-terminal domain that binds to DNA and a changeable C-terminal domain for transcriptional activation [10]. They participate in a diverse array of biological activities, such as growth and development, senescence and reactions to stress [10]. Notably, NAC transcription factors have been well established as key regulators in mediating plant adaptive responses to various abiotic stresses, including drought, high salinity, and cold conditions, underscoring their significant potential as prime targets for stress-tolerance genetic engineering in crops [11].

The NAC family ranks among the largest plant-specific transcription factor families. In different plant species, more than 100 members of this family have been identified [10,11]. In recent years, significant progress has been made in elucidating the roles of NAC transcription factors in stress responses. Studies in Arabidopsis have shown that *NAC* genes such as *ANAC019*, *ANAC055*, and *ANAC072* are induced by salt stress and participate in the regulation of stress-responsive genes [12]. Similarly, in rice (*Oryza sativa* L.), the *NAC* genes *OsNAC2*, *OsNAC5*, and *OsNAC6* have been demonstrated to confer tolerance to multiple abiotic stresses, including salinity and drought [13,14,15,16]. These results emphasize the consistent function of NAC transcription factors in stress responses among different plant species. In soybeans, the NAC family has also been a subject of extensive research, with several members identified as potential regulators of stress responses. For example, studies have shown that *GmNAC06*, *GmNAC11*, *GmNAC085*, and *GmNAC109* can be induced by salt stress. Moreover, when these genes are overexpressed in transgenic plants, they can enhance the plants’ salt tolerance [17,18,19,20]. Additionally, *GmNAC3*, *GmNAC8*, *GmNAC12*, and *GmNAC19* have been implicated in the regulation of drought tolerance [21,22,23,24], further underscoring the multifaceted roles of NAC transcription factors in soybean stress responses. Despite these advancements, the functional characterization of *NAC* genes in soybeans remains incomplete, and the molecular mechanisms by which they contribute to salt tolerance are not yet fully understood.

This study focused on screening the *GmNAC* family genes in soybeans that are potentially associated with the salt stress response. We found that the expression of the *GmNAC035* gene witnessed a sharp rise within an hour after salt treatment. This notable increase strongly suggests that *GmNAC035* is likely to be a key player in how soybeans respond to salt stress. Building on this discovery, we carried out further research to clarify the specific role of *GmNAC035* in enhancing plants’ tolerance to salt. GmNAC035 is a nuclear-localized transcriptional activator. When overexpressed in Arabidopsis, it can enhance the plant’s salt tolerance, reduce oxidative damage, and promote the expression of stress-responsive genes. *GmNAC035* holds the potential to bolster a plant’s resilience to salt stress through the modulation of pivotal stress-response pathways. These pathways encompass the salt overly sensitive (SOS) signaling cascade, the calcium-mediated signaling network, and the abscisic acid (ABA) signaling pathway. The discoveries from this research point out that *GmNAC035* holds promise as an option for genetic engineering. It has the ability to boost the salt tolerance of crops and offer precious candidate gene resources for the development of salt-tolerant soybean varieties.

## 2. Results

### 2.1. Expression Patterns of Predicted GmNAC Genes in Roots Under Salt Stress Conditions

A total of 152 *GmNAC* TFs were identified in earlier studies through a complete genome-wide survey [25]. More recently, studies have identified 180 *GmNAC* genes in the soybean genome Wm82.a2.v1, revealing 32 new *GmNAC* genes [26,27]. Given the increasingly sophisticated development of whole-genome sequencing technology, we conducted a re-identification of the *GmNAC* family genes in the most recent Wm82.a4.v1 version of the soybean genome. The NAC domain was used as the prototype sequence to search against all the deduced protein sequences from the soybean genome. A total of 173 *GmNAC* genes were identified (Appendix A). Compared with the previous version, five new *GmNAC* genes were discovered. To maintain consistency in scientific nomenclature, we adopted a naming system that assigns numbers after the *GmNAC* prefix based on chromosome order and locus position, consistent with previous studies [25]. The newly identified *GmNAC* genes were numbered sequentially from *GmNAC184* onward (Appendix A), maintaining alignment with the naming conventions of the previously described 180 *GmNAC* genes.

Given the pivotal role of the *NAC* genes in plant development and stress responses, we further investigated its members involved in salt stress responses to identify potential candidate genes for enhancing soybean salt tolerance. Utilizing published RNA-seq databases (Appendix A), we analyzed the expression patterns of the *GmNAC* gene family at 1, 2, 4, and 24 h post salt treatment. The results revealed that 101 genes were upregulated following salt treatment, underscoring the broad involvement of *GmNAC* genes in the soybean salt stress response (Appendix A). In this study, we notably focused on genes that were largely induced within 1 h of salt treatment, as these genes are likely to play a critical role in the early stages of stress response, facilitating rapid salt signal perception and transduction. Our analysis identified 48 *GmNAC* genes that were significantly upregulated at 1 h post-treatment, including *GmNAC085* and *GmNAC109*, which have been previously reported, thereby validating the reliability of our findings (Figure 1A). To further elucidate the potential developmental functions of these *GmNAC* genes, we analyzed the expression profiles of the 48 upregulated genes across various developmental stages and tissues using transcriptomic data from the Phytozome database (Appendix A). Given that roots serve as the primary organs for sensing soil stress and are essential for anchoring, nutrient and water absorption, storage, and transport, we examined the expression of these genes in particular in root tissues. The results indicated that 18 of the 48 *GmNAC* genes exhibited high expression levels in soybean roots (Figure 1B).

To further confirm the expression patterns of these 18 *GmNAC* genes under salt treatment, we employed qRT-PCR with gene-specific primers to analyze the expression levels of 18 *GmNAC* genes predicted to be associated with salt stress at 0.5, 1, and 2 h post-treatment (Figure 2). The results indicated that the expression levels of all 18 genes increased to varying degrees following salt treatment. We observed that the expression level of most *GmNAC* genes increased with prolonged treatment time. Interestingly, the expression levels of *GmNAC021*, *GmNAC035*, *GmNAC036*, and *GmNAC088* rapidly increased at 1 h post-salt treatment but decreased at 2 h, suggesting that these genes may play significant roles in the early stages of salt stress response. In this study, we selected *GmNAC035* as the focus for our subsequent research since *GmNAC035* responds to salt relatively more strongly and rapidly.

### 2.2. The Expression of GmNAC035 Is Induced by Hormonal Signals, and It Encodes a Nuclear-Localized Protein with Self-Transactivation Activity

To investigate the function of the *GmNAC035* gene, we initially examined its spatio-temporal expression patterns by collecting samples from various soybean tissue parts. The findings indicated that *GmNAC035* displayed a constitutive expression pattern, with the highest expression level observed in leaves and roots (Figure 3A). By analyzing the expression patterns of *GmNAC* family genes after NaCl treatment, we discovered that *GmNAC035* could respond rapidly to salt stress. Prior research has demonstrated that plant hormones play a vital role in mitigating the detrimental phenotypes induced by abiotic stress in plants and enhancing plant growth adaptation [28]. To determine whether *GmNAC035* responds to hormone signals, we treated soybean leaves with various hormones and assessed the expression levels of *GmNAC035* at different time points post-treatment. Following abscisic acid (ABA) treatment, the expression level of *GmNAC035* increased continuously and significantly, reaching over 20-fold that of the control at 48 h (Figure 3B). The expression level of *GmNAC035* peaked 24 h after methyl jasmonate (MeJA) treatment and was approximately eight-fold that of the control (Figure 3C). The prompt response of *GmNAC035* to hormone treatment strongly implied that it may have a specific function in the hormone signal transduction pathway during the salt stress response.

Previous reports have indicated that *NAC* family genes belong to a class of nuclear-localized transcription factors with transcriptional activation activity [29]. To confirm whether GmNAC035 participates in the transcriptional activation process, we inserted the full-length *GmNAC035* gene into the pGBKT7 vector and introduced it into the yeast strain AH109. The negative control, positive control, and BD-*GmNAC035* transformants all exhibited favorable growth on synthetic defined (SD)/-Trp medium. Nevertheless, only the yeast cells of the positive control and BD-*GmNAC035* were able to survive on SD/-Trp/-His/-Leu medium (Figure 3D). To further analyze the subcellular localization of GmNAC035, we constructed a *GmNAC035-GFP* fusion vector. Subsequently, the empty *GFP* vector and the *GmNAC035-GFP* fusion vector were transiently transformed into tobacco leaf epidermal cells. Using laser confocal microscopy, we observed that the fluorescence signal of the empty GFP vector was distributed across the cytoplasm, nucleus, and cell membrane. Conversely, the fluorescence signal of the GmNAC035-GFP fusion protein colocalized with the nuclear marker protein and was exclusively located in the nucleus (Figure 3E). These findings suggest that GmNAC035 is a transcription factor that is localized in the nucleus and participates in transcriptional activation.

### 2.3. The Overexpression of GmNAC035 Improves Salt Stresses Tolerance of Arabidopsis

To further validate the role of *GmNAC035* in responding to salt stress, we generated transgenic *Arabidopsis thaliana* lines overexpressing *GmNAC035*. After obtaining homozygous and stable Arabidopsis overexpression lines, we conducted an expression analysis on three of these lines using RT-qPCR. The results indicated that the expression level of *GmNAC035* was significantly elevated (Figure 4B). Simultaneously, we exposed wild-type Arabidopsis and the *GmNAC035*-overexpressing lines to different concentrations of NaCl (0 mM, 75 mM, 100 mM, and 125 mM) on 1/2 MS for 6 days. The results demonstrated that there was no significant disparity between the two types of lines under normal conditions (Figure 4A). However, when exposed to 100 mM and 125 mM salt concentrations, the fresh weight of the *GmNAC035*-overexpressing lines was significantly higher than that of the wild type (Figure 4C). Although there was no significant difference in root length between the overexpressing Arabidopsis lines and the wild type (Appendix A), the number of lateral roots in the overexpressing lines was obviously greater than that in the wild type (Figure 4D). Additionally, we determined the survival rates of different Arabidopsis lines when exposed to a 100 mM salt concentration. The results revealed that the survival rate of the overexpressing lines was notably greater than that of the wild type (Figure 4E). These results provided evidence that *GmNAC035* positively regulates salt resistance in Arabidopsis.

### 2.4. GmNAC035 Contributed to the Elimination of ROS

An increasing body of evidence indicates that salt stress triggers the rapid generation and accumulation of ROS, which impose toxic impacts on plants by inducing oxidative stress [30]. Hydrogen peroxide (H_2_O_2_) and superoxide anions (O_2_^−^) are the main components of ROS in plants. 3,3′-Diaminobenzidine (DAB) and nitroblue tetrazolium (NBT) can be used to detect H_2_O_2_ and O_2_^−^, respectively. DAB reacts with H_2_O_2_ to form a brown precipitate, while NBT reacts with O_2_^−^ to produce a blue formazan precipitate. To ascertain whether *GmNAC035* participates in the ROS accumulation process, we stained Arabidopsis plants treated with 0 mM and 100 mM NaCl using DAB and NBT, respectively. The results demonstrated that in the absence of salt stress, there were negligible differences in ROS accumulation among different Arabidopsis lines. However, when subjected to 100 mM NaCl, compared with the control, the ROS accumulation level in the *GmNAC035*-overexpressing Arabidopsis lines was relatively lower (Figure 5A,B), which was consistent with the phenotype after salt stress. Malondialdehyde (MDA), which results from peroxidation reactions, acts as an indicator for gauging the extent of plasma membrane impairment and the intensity of stress-induced responses. To evaluate whether *GmNAC035* is involved in the oxidative damage process, we measured the MDA concentration (Appendix A). In normal growth circumstances, there was no significant disparity in MDA content between the control plants and the *GmNAC035*-overexpressing lines. However, after exposure to 100 mM NaCl, the content of MDA in the *GmNAC035*-overexpressing lines was prominently lower than that in the wild type (Figure 5C). This finding suggested that the *GmNAC035*-overexpressing lines endured less oxidative damage under salt stress. In order to maintain the redox homeostasis of ROS and resist the harmful effects caused by ROS and biotic/abiotic stresses, plants have developed scavenging mechanisms [31]. These mechanisms make use of enzymatic components and non-enzymatic antioxidants, such as superoxide dismutase (SOD) and peroxidase (POD) [32]. Subsequently, we measured the activities of SOD and POD in different Arabidopsis lines under salt stress (Appendix A). We found that without exposure to salt, the SOD activity in the *GmNAC035*-overexpressing Arabidopsis lines was significantly higher than that in the wild type. Upon exposure to 100 mM NaCl, the SOD activity in the overexpressing lines was markedly lower than that in the wild type (Figure 5D). In contrast, the POD activity in the overexpressing lines was consistently significantly higher than that in the wild type (Figure 5E). These results imply that POD in the *GmNAC035*-overexpressing lines plays a more crucial role in alleviating oxidative damage.

### 2.5. Overexpression of GmNAC035 Increases Expression of Genes Related to Abiotic Stress Resistance

Throughout the evolutionary process, plants have evolved diverse adaptive strategies to contend with high-salt stress [6]. To elucidate the potential molecular mechanism underpinning the enhanced salt tolerance of plants overexpressing the *GmNAC035* gene, we investigated the transcriptional abundance of abiotic stress-related genes under salt stress conditions (Appendix A). The salt overly sensitive (SOS) pathway assumes a pivotal and indispensable role in endowing plants with the capacity to endure salt stress. This pathway encompasses three key proteins, namely SOS1, SOS2, and SOS3 [30]. The cytoplasmic calcium signal induced by salt stress is sensed by SOS3, a calcium-binding protein with an EF-hand structure. Subsequently, SOS3 interacts with and activates the serine/threonine protein kinase SOS2. The activated SOS2 phosphorylates SOS1, thereby promoting its activity. SOS1, a plasma membrane-localized Na^+^/H^+^ antiporter located downstream in the SOS pathway, enhances plant salt tolerance by facilitating the efflux of excess intracellular Na^+^ and regulating the long-distance transport of Na^+^ in plants [6,33,34]. In plants, the activity of SOS1 is strictly regulated: it is activated under high-salt conditions and inactivated through various regulatory mechanisms under low-salt conditions to achieve intracellular sodium homeostasis. Our results indicated that, despite the absence of a significant difference in the expression level of *AtSOS1* in the *GmNAC035*-overexpressing Arabidopsis lines compared to the control under salt stress conditions, the expression levels of *AtSOS3* and *AtSOS2* were significantly higher than those in the control (Figure 6A–C). This is advantageous for promoting the activity of AtSOS1 and consequently enhancing the efflux of excess sodium ions in the *GmNAC035*-overexpressing lines. Moreover, NHX1 plays a crucial role in maintaining Na^+^/K^+^ homeostasis within the vacuole under salt stress conditions, and overexpression of *NHX1* exerts a positive influence on plant salt tolerance [6,35]. We also found that the expression level of *AtNHX1* in the *GmNAC035*-overexpressing Arabidopsis lines was almost twice as high as that in the control (Figure 6D).

Under salt stress conditions, plant cells are capable of sensing high concentrations of sodium ions and initiating calcium influx, resulting in a rapid and transient elevation in the cytoplasmic calcium concentration. This directly modulates the transcription levels of genes involved in the salt stress response. This process needs to be interpreted by different calcium sensors, including calmodulin (CaM), calcium-dependent protein kinases (CDPKs), calmodulin-like proteins (CMLs), and calcineurin B-like protein/calcineurin B-like protein-interacting protein kinase (CBL/CIPK) [36]. Our findings demonstrated that the expression levels of two calcium sensors, *AtCDPK6* [37] and *AtCBL1* [38], which positively govern plant salt tolerance, were significantly higher in the *GmNAC035*-overexpressing lines than in the control after salt stress treatment (Figure 6E,F).

ABA is the principal phytohormone accountable for salt and osmotic stress signaling in plant cells [6,33]. The findings of our research suggested that the expression levels of *AtABA3* [39], which is associated with ABA biosynthesis; *AtRD22* [40,41], an ABA-mediated osmotic stress response factor; and *AtPPRT3* [42], which positively governs ABA signaling, were also significantly higher in the *GmNAC035*-overexpressing lines than in the control after salt stress treatment (Figure 6G–I). These results suggest that *GmNAC035* may positively regulate plant tolerance to salt stress by modulating the expression levels of key genes related to salt tolerance regulation and genes in the ABA signaling pathway.

## 3. Discussion

Soybean (*Glycine max*), as a typical salt-sensitive crop, is susceptible to the negative effects of salt stress at all stages of growth and development [43]. Salt stress not only significantly reduces soybean yield but also adversely affects its quality. Therefore, breeding salt-tolerant soybean varieties has become an important goal in current soybean breeding. A new approach to improving the efficiency of selecting salt-tolerant varieties is provided by the application of molecular breeding technology. Among plant-specific transcription factor families, NAC proteins, which are one of the largest, are commonly found in different terrestrial plants [44]. Research indicates that the NAC transcription factor family exerts a vital regulatory influence on various aspects of plants, including their growth process, development stages, and how they respond to stress [11]. In this study, we conducted a genome-wide analysis based on the latest version of the soybean genome (Wm82.a4.v1) and identified a total of 173 *GmNAC* genes, including five newly discovered gene members. To screen for potential salt-tolerant *GmNAC* genes, we systematically analyzed RNA-Seq databases under salt stress conditions and found that 101 *GmNAC* genes were upregulated under salt stress, with 48 genes being significantly induced within one hour of salt treatment. Considering that roots are the primary organs for sensing soil stress and the first line of defense against stress, we combined tissue-specific expression databases to screen out 18 *GmNAC* genes mainly expressed in roots. It was confirmed through RT-qPCR validation that salt stress truly and significantly induced the expression levels of these genes. Notably, *GmNAC021*, *GmNAC035*, *GmNAC036*, and *GmNAC088* showed a transient but significant increase in expression within one hour after salt treatment, suggesting their potential involvement in the early signaling cascade of salt stress response.

Based on these findings, we selected *GmNAC035* for further investigation. *GmNAC035* has the highest expression levels in leaves and roots. The expression of this gene responds to both ABA and MeJA treatments, indicating its potential involvement in the ABA and JA signaling pathways, both of which play important roles in plant stress responses [45]. Yeast transcription activation experiments confirmed that GmNAC035 possesses typical transcription activation functions. Subcellular localization studies using GFP fusion proteins revealed that GmNAC035 is specifically localized in the nucleus, consistent with the characteristics of NAC family proteins as nuclear-localized transcription factors [44]. To elucidate the salt tolerance function of *GmNAC035*, we constructed transgenic Arabidopsis lines overexpressing *GmNAC035*. Under salt stress conditions, the transgenic plants exhibited significantly enhanced salt tolerance, manifested as increased biomass and improved survival rates. Meanwhile, we also noticed that the *GmNAC035*-overexpressing transgenic Arabidopsis lines had significantly more lateral roots than the wild type, but there was no significant difference in root length. Since the development of lateral roots plays a vital role in plants’ adaptation to the environment and their ability to cope with abiotic stresses, this change in root system architecture is likely to affect the plant’s tolerance to high salt and drought [20].

Salt stress has the potential to elevate the production of ROS and change the redox state within cells. This alteration can cause DNA damage, and in the end, give rise to oxidative stress, which exerts a negative impact on plant growth [46]. We subsequently investigated the physiological mechanisms by which *GmNAC035* confers salt tolerance to plants. Findings revealed a reduced accumulation of reactive ROS and MDA content in the *GmNAC035*-overexpressing transgenic Arabidopsis lines after salt stress, indicating that overexpression of *GmNAC035* effectively alleviates oxidative damage caused by salt stress. SOD catalyzes the dismutation of O_2_^−^ to H_2_O_2_, which is then decomposed by POD and CAT enzymes into H_2_O and O_2_ [47]. In our study, prior to salt treatment, *GmNAC035*-overexpressing lines displayed markedly higher SOD and POD activities compared to wild-type plants, demonstrating that *GmNAC035* overexpression enhances the antioxidant system. However, under salt stress conditions, while POD activity remained significantly elevated in the overexpression lines, SOD activity failed to increase further, implying a more pivotal role for POD in combating oxidative damage in these transgenic plants. Given that the antioxidant machinery in higher plants involves both enzymatic and non-enzymatic components [31], it is reasonable to speculate that non-enzymatic antioxidants may also contribute substantially to the observed mitigation of oxidative stress in these overexpression lines.

It is well known that, when exposed to saline environments, plants initiate intricate salt tolerance mechanisms that encompass multiple genetic pathways. The *GmNAC* transcription factors likely confer enhanced stress resistance through their regulatory influence on downstream stress-responsive gene networks. In this study, further research revealed that the expression of several key salt stress response genes (*AtSOS2*, *AtSOS3* and *AtNHX1*) was significantly upregulated in transgenic plants. Research has shown that the SOS pathway, which consists of SOS1, SOS2, and SOS3, is essential for sustaining ion homeostasis when plants are exposed to salt stress [48]. NHXs are putative Na^+^/H^+^ exchangers that are responsible for transporting Na^+^ from the cytoplasm into the vacuole. During salt stress, the SOS3–SOS2 complex exerts a positive regulatory effect on NHX activity [49]. The upregulation of *AtSOS2* and *AtSOS3* suggests that *GmNAC035* may enhance salt tolerance by activating the SOS and NHX pathway, promoting sodium ion efflux and maintaining intracellular ion balance. CDPKs and CBLs, as important calcium signal sensors, play key roles in stress signal transduction [50]. In our study, the expression levels of *AtCDPK6* and *AtCBL1* were significantly increased in transgenic plants, highlighting the important role of calcium signaling in *GmNAC035*-mediated salt tolerance. Notably, under salt stress conditions, the expression of ABA biosynthesis and signaling-related genes (*AtABA3*, *AtRD22*, and *AtPPRT3*) was significantly upregulated in transgenic plants, further confirming *GmNAC035*’s involvement in ABA-mediated stress responses. As a pivotal hormone regulating plant responses to abiotic stress, the upregulation of genes associated with ABA suggests that *GmNAC035* may enhance plant adaptability to osmotic stress by modulating the ABA signaling pathway.

This study systematically elucidated the multiple regulatory roles of *GmNAC035* in plant salt tolerance, providing new insights into the molecular mechanisms of NAC transcription factors in plant stress responses and offering valuable candidate gene resources for soybean salt-tolerant breeding. Future research should be predominantly concentrated on conducting genetic transformation of *GmNAC035* in soybeans to evaluate its potential impact on soybean yield and quality.

## 4. Materials and Methods

### 4.1. Identification of NAC Genes in the Soybean Genome and Analysis of Tissue-Specific Expression

The Phytozome database (https://phytozome-next.jgi.doe.gov/, accessed on 14 September 2023) was used to download all protein sequences of soybean (version: Glycine max Wm82.a4.v1) [51]. The HMM model file (PF02365) for the NAM domain was retrieved from the Pfam database (http://pfam.xfam.org/, accessed on 14 September 2023) [52]. Using the HMMsearch tool (version 3.0) with default parameters [53], sequences containing the NAM domain were screened from the complete protein sequences of soybean and Arabidopsis. Newly identified GmNAC family genes were named in accordance with previous studies (Appendix A) [25,26].

Based on the RNA-Seq database [54], the expression levels of genes belonging to the *GmNAC* family in the roots of soybean under salt stress conditions were analyzed, and heatmaps depicting *GmNAC* gene expression under salt stress were generated using TBtools-II (version 2.142) software. Expression data of GmNAC genes upregulated under salt stress across various tissues were extracted from the soybean tissue expression database (Glycine max Wm82.a4) using Blast (https://phytozome-next.jgi.doe.gov/, accessed on 23 September 2023). Subsequently, tissue-specific expression heatmaps of GmNAC genes were constructed using TBtools.

### 4.2. Plant Materials, Growth Conditions, and Stress Treatments

The soybean cultivar Tianlong No. 1 [55] was used as the wild type. Soybean seeds were germinated for 6 days on vermiculite and then transferred to 1/2 Hoagland nutrient solution (Macklin, Shanghai, China) for growth in the plant growth chamber under a 16 h (28 °C) light/8 h (22 °C) dark photoperiod. The relative humidity was maintained at 60%. Quantities of 200 mM NaCl, 150 µM ABA, and 100 mM MeJA, which were prepared using 1/2 Hoagland nutrient solution, were applied to treat the plants when the first trifoliate leaves were fully expanded [22,56,57]. Samples were collected at 0 h, 6 h, 12 h, 24 h, and 48 h after treatment for subsequent RNA extraction.

### 4.3. RNA Extraction and RT-qPCR Analysis

Total RNA was extracted from soybean and Arabidopsis using the Steady Pure Universal RNA Extraction Kit (Accurate Biotechnology, Hunan, China). According to the manufacturer’s protocol, TB Green Premix Ex Taq^TM^ II (Takara, Tokyo, Japan) was used for RT-qPCR analysis of the cDNA, which was detected via the fully automatic medical PCR analysis system (Gentier 96R, TIANLONG, Shaanxi, China). The specific primers used in this experiment are listed in Appendix A. The transcriptional levels of the target genes were analyzed with GmCons4 and AtActin2 serving as internal reference genes [58]. The RT-qPCR data were analyzed using the 2^−ΔCT^ method. The data presented here are representative of two biological replicates. Each sample underwent analysis with three technical replicates, and Excel 2016 was employed to assess statistical significance.

### 4.4. Subcellular Localization of GmNAC035

The full-length CDS sequence of *GmNAC035* was amplified from Tianlong No. 1 using the KOD FX (TOYOBO, Osaka, Japan), following the manufacturer’s instructions. The pCAMBIA1305-GFP plasmid with the 35S promoter and the NOS Terminator was digested with QuickCut Spe I (Takara, Tokyo, Japan), and then the digested plasmid and the GmNAC035 gene fragment excluding the stop codon were ligated using the In-Fusion snap assembly master mix (Takara, Tokyo, Japan) [59,60]. The positive recombinant plasmid, pCAMBIA1305-GmNAC035-GFP (Appendix A), was transformed into Escherichia coli DH5α strain (YEASON, Shanghai, China) according to the manufacturer’s instructions. Subsequently, the positive plasmid was introduced into the *Agrobacterium tumefaciens* GV3101 strain [61] and transiently expressed in tobacco leaves, as described previously [62]. Leaves were collected 48 h after transformation. OsD53-mCherry was used as a nuclear marker and came from Nanjing Agricultural University [63]. The empty vector served as a negative control. Observations were made using a laser scanning confocal microscope (LSM980, Carl Zeiss, Oberkochen, Germany). Regarding GFP signals, the excitation wavelength was configured at 488 nm, while the emission wavelength spanned from 505 to 530 nm. When it came to mCherry signals, an excitation wavelength of 587 nm was employed, and the emission wavelengths were measured within the 600–630 nm interval.

### 4.5. Transcriptional Self-Activation Activity Analysis

The pGBKT7 plasmid (Clontech, CA, USA) was digested by QuickCut *Eco*R I (Takara, Tokyo, Japan), and then the full-length CDS sequence of the *GmNAC035* gene was recombined with the linearized pGBKT7 plasmid using the In-Fusion snap assembly master mix (Takara, Tokyo, Japan). Following the manufacturer’s instructions, the positive recombinant plasmid, pGBKT7-GmNAC035, was transformed into the yeast strain AH109 (YEASON, Shanghai, China). The pGBKT7-GmNAC181 plasmid was used as a positive control [64], and the empty pGBKT7 vector served as a negative control. The growth of positive colonies on SD/-Trp medium confirmed the presence of the transgene, while the growth on SD/-His/-Leu/-Trp medium indicated transcriptional activation activity. The information regarding the pGBKT7 vector is available at Clontech (https://www.takarabio.com/documents/Vector%20Documents/pGBKT7%20Vector%20Information.pdf, accessed on 10 January 2024).

### 4.6. Plant Transformation and Screening of Transgenic Plants

To obtain Arabidopsis thaliana plants overexpressing GmNAC035, we used the pCAMBIA 1300 plant expression vector containing the constitutive promoter pUBQ10 [65]. The pCAMBIA1300-GmNAC035 plasmid (Appendix A) was introduced into the Agrobacterium tumefaciens GV3101 strain [61]. The floral dip method was performed as described previously to transform the wild-type (WT) Arabidopsis Col 0 plants. [66]. Seeds harvested from transformed plants were screened for hygromycin resistance (40 mg/L). The expression levels of GmNAC035 in T_3_ generation lines were detected using RT-qPCR, with specific primers listed in Appendix A. Homozygous T_3_ plants were used for further analysis.

### 4.7. Salt Tolerance Analysis of GmNAC035 Overexpression Lines

Seeds of three GmNAC035 overexpression transgenic lines and WT Arabidopsis were sterilized and incubated at 4 °C for 3 days, followed by cultivation on 1/2 MS medium for 4 days. The medium was prepared by mixing 2.22 g/L of PhytoTech M519 Murashige & Skoog Basal Medium with Vitamins (PhytoTech, Lenexa, KS, USA), 5 g/L of Gelzan™ CM (Sigma, Saint Louis, MO, USA), and 10 g/L of sucrose (Sigma, Saint Louis, MO, USA). After the mixture was prepared, the pH value was adjusted to 5.7–5.8 using 1 M KOH. The seedlings were then transferred to 1/2 MS medium mixed with different concentrations of NaCl (0 mM, 75 mM, 100 mM, and 125 mM) for salt stress treatment for 6 days. Seedlings were photographed, and their fresh weight, lateral root number, and survival rate under 100 mM NaCl stress were measured. MDA content and the activities of SOD and POD were determined using 100 mg of leaf tissue. The assays were performed using the Malondialdehyde (MDA) Content Assay Kit (Beijing Solarbio Science & Technology, Beijing, China), Peroxidase (POD) Activity Assay Kit (Beijing Solarbio Science & Technology, Beijing, China), and Superoxide Dismutase (SOD) Activity Assay Kit (Beijing Solarbio Science & Technology, Beijing, China), following the manufacturer’s protocol. Under salt stress, plants produce hydrogen peroxide (H_2_O_2_) and superoxide (O_2_^−^), which mediate various physiological and biochemical processes. H_2_O_2_ and O_2_^−^ were detected using 3, 3′-diaminobenzidine (DAB) (Shanghai yuanye Bio-Technology, Shanghai, China) and nitroblue tetrazolium (NBT) (Shanghai yuanye Bio-Technology, Shanghai, China), respectively. Leaves were immersed in 5 mg/mL DAB (aqueous solution, pH 3.8) and 0.5 mg/mL NBT (aqueous solution, pH 7.5) in the dark for 20 h. Afterward, the leaves were transferred to 95% ethanol and boiled in a water bath for 15 min. After cooling, the leaves were soaked in 95% ethanol at room temperature until completely decolorized [67].

## 5. Conclusions

In this study, 173 *GmNAC* genes were re-identified in the latest soybean genome database (Wm82.a4.v1), with 101 being salt-induced and 48 hypothesized as being crucial for the early salt stress response. Additionally, 18 *GmNAC* genes highly expressed in roots and rapidly responsive to salt stress were verified. Notably, the expression of *GmNAC035* was rapidly upregulated within one hour after salt treatment. GmNAC035 functions as a nuclear-localized transcriptional activator. Our exploration of the regulatory function of *GmNAC035* in salt tolerance has shown that it plays a positive role in modulating plants’ ability to withstand salt stress by mitigating oxidative damage and fine-tuning the expression of genes associated with the salt stress response, including the SOS pathway, calcium signaling, and ABA signaling. Our findings suggested that *GmNAC035* plays a regulatory role in plant salt tolerance, thereby offering novel genetic resources for enhancing soybean salt tolerance.

## Figures and Tables

**Figure 1 plants-14-01391-f001:**
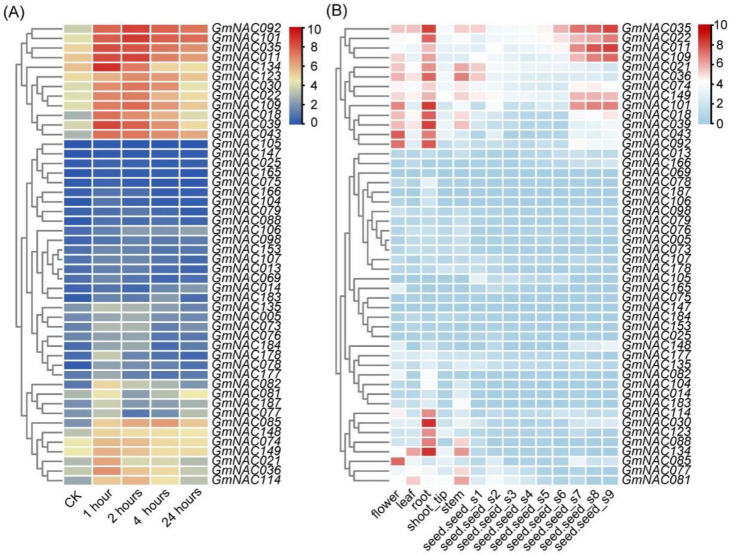
Expression profiles of *GmNAC* genes under NaCl stress and in different tissues. (**A**) Expression profiles of *GmNAC* genes with at least a 2-fold up-regulation in expression after 1 h of treatment with 100 mM NaCl in soybean roots. (**B**) Expression profiles of *GmNAC* genes in different tissues. The RPKM normalized values of the genes were log2-transformed and visualized as a heatmap. The gene IDs are shown in Appendix A.

**Figure 2 plants-14-01391-f002:**
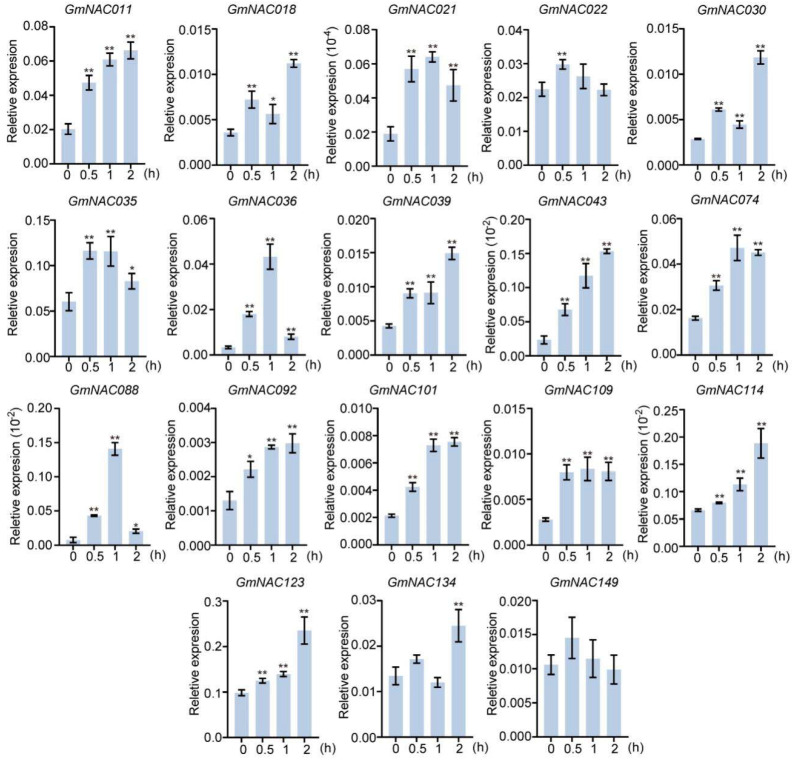
Transcriptional levels of 18 *GmNAC* genes in “Tianlong No. 1” soybean under treatment with 150 mM NaCl. Representative data from two biological replicates are shown. Each sample was analyzed in three technical replicates. The data represent the mean ± SD, * *p* < 0.05, ** *p* < 0.01 (Student’s *t*-test).

**Figure 3 plants-14-01391-f003:**
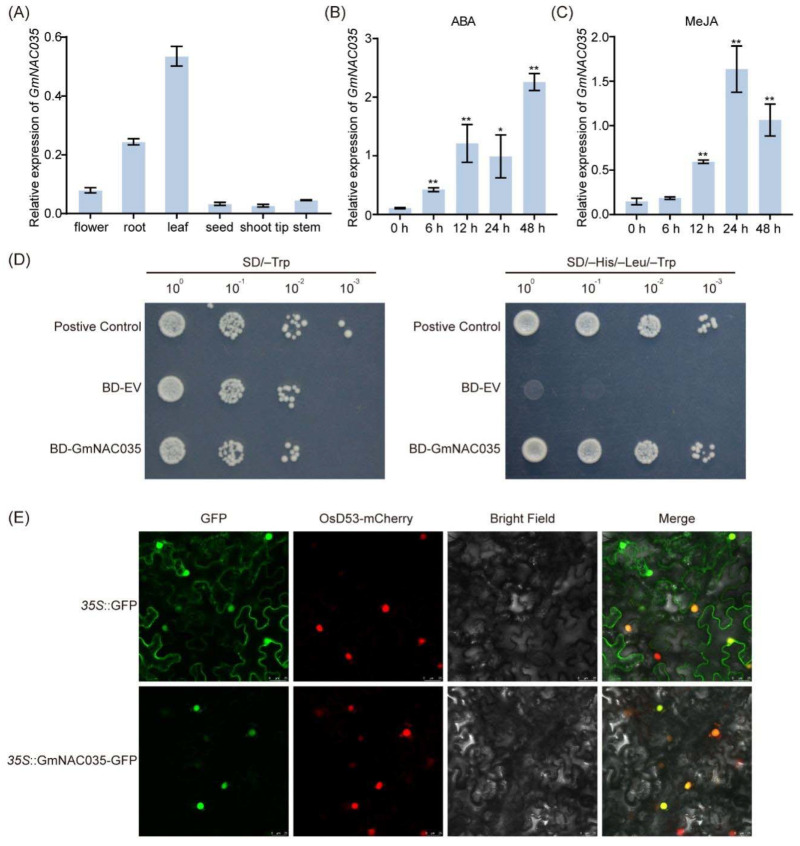
Expression pattern of the *GmNAC035* gene in “Tianlong No. 1” soybean and analysis of the transcriptional activity and subcellular localization of the GmNAC035 protein. (**A**) Transcriptional levels of the *GmNAC035* gene in different tissues. (**B**,**C**) Transcriptional levels of the *GmNAC035* gene under stress of 150 µM ABA and 100 mM MeJA, respectively. Representative data from two biological replicates are shown. Each sample was analyzed in three technical replicates. The data represent the mean ± SD, * *p* < 0.05, ** *p* < 0.01 (Student’s *t*-test). (**D**) Determination of the transcriptional self-activation activity of the GmNAC035 protein in yeast. Yeasts transformed with BD-GmNAC035, negative control (BD-EV), and positive control (BD-GmNAN181) plasmids were grown on SD/-Trp medium and SD/-His/-Leu/-Trp medium. (**E**) Analysis of the subcellular localization of the GmNAC035 protein. *35S::GFP* was used as the control. Scale bar = 25 µm.

**Figure 4 plants-14-01391-f004:**
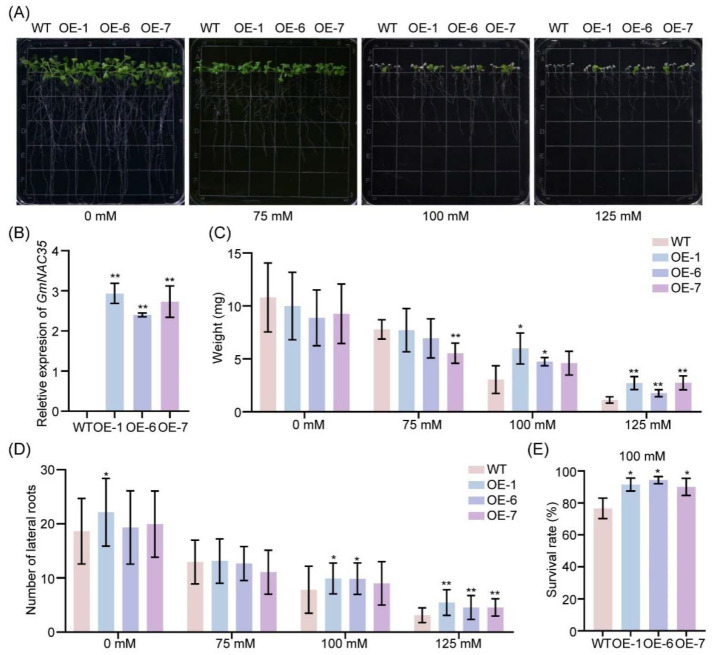
Overexpression of *GmNAC035* increases salt tolerance in Arabidopsis. (**A**) Phenotypic observation of wild-type and *GmNAC035* overexpression Arabidopsis lines under salt stress at different concentrations. (**B**) Transcriptional levels of the *GmNAC035* gene in wild-type Arabidopsis and transgenic lines. RT-qPCR analysis was performed using the 2^−ΔCT^ method. (**C**,**D**) Statistical analysis of the fresh weight (**C**) and the number of lateral roots (**D**) of wild-type Arabidopsis and overexpression lines under salt stress at different concentrations. (**E**) Statistical analysis of the survival rate of overexpression lines under 100 mM salt stress. The data represent the mean ± SD. *n* > 65. * *p* < 0.05, ** *p* < 0.01 (Student’s *t*-test).

**Figure 5 plants-14-01391-f005:**
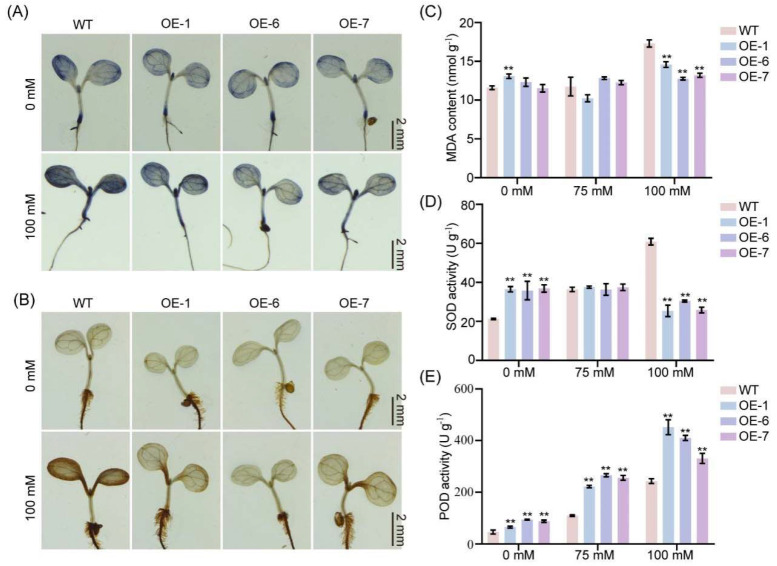
Visualization of ROS and determination of antioxidant enzyme activities. (**A**,**B**) Staining observations of NBT (**A**) and DAB (**B**) in wild-type and *GmNAC035*-overexpressing Arabidopsis lines under 0 mM and 100 mM salt stress. Scale bar = 2 mm. (**C**–**E**) MDA content (**C**), SOD activity (**D**), and POD activity (**E**) in wild-type and *GmNAC035*-overexpressing Arabidopsis lines under 0 mM, 75 mM, and 100 mM salt stress. The data represent the mean ± SD. ** *p* < 0.01 (Student’s *t*-test).

**Figure 6 plants-14-01391-f006:**
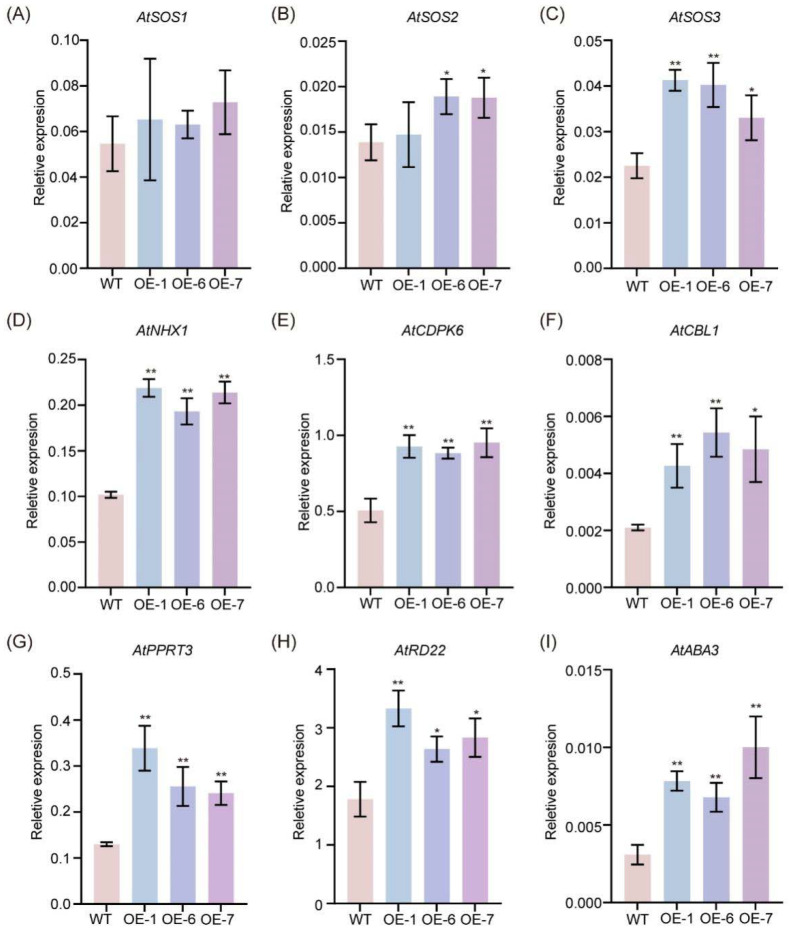
Transcriptional levels of salt tolerance-related genes in wild type and *GmNAC035*-overexpressing Arabidopsis lines, including *AtSOS1/2/3* (**A**–**C**), *AtNHX1* (**D**), *AtCDPK6* (**E**), *AtCBL1* (**F**), *AtPPRT3* (**G**), *AtRD22* (**H**), and *AtABA3* (**I**). Leaves were collected from Arabidopsis seedlings treated with 100 mM NaCl for 6 days. RT-qPCR analysis was performed using the 2^−ΔCT^ method. Representative data from two biological replicates are shown. Each sample was analyzed in three technical replicates. The data represent the mean ± SD. * *p* < 0.05, ** *p* < 0.01 (Student’s *t*-test).

## Data Availability

The original contributions presented in the study are included in the article/Appendix A.

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
