# Peer review of "NAC Transcription Factor GmNAC035 Exerts a Positive Regulatory Role in Enhancing Salt Stress Tolerance in Plants"

_plants, 2025, doi:10.3390/plants14091391_

Round 1

Reviewer 1 Report

Comments and Suggestions for Authors

This study concentrates on the GmNAC035 gene in soybeans and its mechanism in enhancing plant salt stress tolerance. Through a series of experiments, the unique function of this transcription factor in regulating plant salt tolerance is revealed. While prior studies have explored the roles of NAC family genes in salt tolerance, systematic research on GmNAC035 is limited, making this study innovative. Below are some suggestions for improving the article, and I expect to see a detailed response letter indicating specific revisions:

1 Expand on global soil salinization and the importance of soybeans in global agriculture.

2 In lines 31-34, cite the latest literature on salt stress reducing crop yields: https://doi.org/10.1016/j.fcr.2025.109747.

3 Explicitly state the specific objectives and significance of studying the GmNAC035 gene in the introduction, and its potential for improving soybean salt tolerance.

4 In lines 408-409, detail the preparation of solutions and maintenance of salt treatment concentration.

5 Deepen the discussion by analyzing results more thoroughly, exploring their implications and mechanisms, and comparing them with previous studies.

6 Examine how GmNAC035 interacts with other salt tolerance genes or signaling pathways and its specific mechanism in enhancing soybean salt tolerance.

7 Acknowledge the study's limitations and suggest future research directions in the discussion.

8 Concisely summarize key findings and viewpoints in the conclusion.

Author Response

Response to Reviewer 1:

We acknowledge the reviewer1’s helpful comments and recommendations, which are valuable for improving our manuscript.

  1. Expand on global soil salinization and the importance of soybeans in global agriculture.

Reply: Thank you for your comments. We added information about global soil salinization and the importance of soybeans in global agriculture in the introduction (New Line 29-40).

  1. In lines 31-34, cite the latest literature on salt stress reducing crop yields: https://doi.org/10.1016/j.fcr.2025.109747.

Reply: Thank you for your comment. We cited the latest literature on how salt stress reduces crop yields (New Reference 2 ).

  1. Explicitly state the specific objectives and significance of studying the GmNAC035 gene in the introduction, and its potential for improving soybean salt tolerance.

Reply: Thank you for your valuable comments. As you suggested, in the introduction, we described the purpose and significance of the research on the GmNAC035 gene, as well as the potential of this gene to enhance the salt tolerance of soybeans (New Line 81-86 and 93-95).

  1. In lines 408-409, detail the preparation of solutions and maintenance of salt treatment concentration.

Reply: Thank you for your comments. We supplemented the materials and methods. The 1/2 Hoagland nutrient solution was purchased from Macklin Company. 200 mM NaCl, 150 µM ABA, and 100 mM MeJA, which were prepared using 1/2 Hoagland nutrient solution, were applied to the plants when the first trifoliate leaves were fully expanded (New Line 433-436).

  1. Deepen the discussion by analyzing results more thoroughly, exploring their implications and mechanisms, and comparing them with previous studies.

Reply: Thank you for your comments. We supplemented and revised the discussion section.

  1. Examine how GmNAC035 interacts with other salt tolerance genes or signaling pathways and its specific mechanism in enhancing soybean salt tolerance.

Reply: Thank you very much for your comments. The GmNAC035 likely confers enhanced stress resistance through its regulatory influence on downstream stress- responsive gene networks, including the SOS pathway, calcium signaling, and ABA signaling (New Line 384-405).

  1. Acknowledge the study's limitations and suggest future research directions in the discussion.

Reply: Thank you for your comments.This study systematically elucidated the important regulatory role of GmNAC035 in the salt tolerance of Arabidopsis thaliana, and research on soybeans is currently underway. Future research should be predominantly concentrated on conducting genetic transformation of GmNAC035 in soybeans to evaluate its potential impact on soybean yield and quality (New Line 406-411).  

  1. Concisely summarize key findings and viewpoints in the conclusion.

Reply: Thank you for your comments. We have concisely summarized the main findings and viewpoints in the conclusion.

Reviewer 2 Report

Comments and Suggestions for Authors

Synopsis:

The manuscript re-identified NAC genes in a newer version of the soybean genome. This analysis of the Wm82.a4.v1 genome identified 5 new NAC genes that were not present in the Wm82.a2.v1 assembly.  In total, 152 NAC genes were identified in the Wm82.a2.v1 assembly whereas 173 NAC genes were identified in this study using the Wm82.a4.v1 assembly.  The NAC gene were named according to a previous study with the new genes getting new names in sequence. The authors utilized published RNA-seq datasets to analyze the expression of the NAC genes at 1, 2, 4 and 24 hrs of salt treatment. This indicated that 101 of the 173 identified NAC genes were up-regulated during salt stress. 48 NAC genes were strongly up-regulated in the first hour of salt stress.  Two of those genes (GmNAC085 and GmNAC109) have previously been reported as being up-regulated due to salt stress. Utilizing expression data from Phytozome database, it was observed that 18 of the 48 up-regulated genes had significantly higher expression under salt stress (Fig. 1).  18 NAC genes were used in qRT-PCR to quantify their expression levels under salt stress.  All 18 had an increase in cDNA amount of varying degrees. Four genes were rapidly induced by salt and were chosen for further study.  GmNAC035 was eventually chosen for further study based on early and robust expression in root tissue (Fig. 2). Utilizing a GmNAC035 GFP construct, the authors showed that the fluorescence was predominantly located in the nucleus where you would expect a transcription factor to be localized (Fig.3E). Arabidopsis over expressing lines were created (Fig. 4A)  and three Over Expressing lines (OE) were challenged with media with varying concentrations of salt. There was no difference in early growth of the OE lines and controls in normal conditions but when grown on media containing salt there was a difference in fresh weight, lateral root growth and survival. There was also more cell death and membrane peroxidation in wild-type  than in OE lines (Fig. 5).  Over expression of GmNAC035 in Arabidopsis plants induced over expression of selected Arabidopsis genes associated with salt tolerance (Fig. 6) supporting the notion that GmNAC035 functions as a transcription regulator of other salt tolerance genes.

Issues:

Lines 89-91 are confusing. Could you re-write them to make the point easier to understand.

Line 163 Why was pGBKT7 chosen? A diagram of the plasmid and insertion would make the text easier to understand.  What procedure was used?  Cite the procedure. What procedure was used to clone it into the yeast strain AH109?

Line 168 How was GmNAC35-GFP made? Cite the paper. A diagram of the construct would help understanding it. What was used as promoter? What was used as a stop signal? What was the MCS?

Line 168-170 How was the GmNAC035-GFP construct transformed into tobacco cells?  Cite the procedure.  Where did the mCherry  used in Fig. 3 E come from?

Line 193 What is MS medium? If you made it from scratch you have to describe it.  If it was a commercial product then cite the product name. You need to describe how the salt was added to the medium.  Was it added during the preparation or after the preparation (flooded onto the medium).

Line 220 Why use NBT and DAB?  What do they do? Which is a cell viability stain and which is an indicator of lipid peroxidation.

Fig.5 panels C, D and E.  Bar charts are a summary of the data.  There should be a table in supplementary data with that data.

Fig 6. Bar chart data should be in the supplementary files

Line 393 what HMMsearch tool options were used if any

Line 405 what was the germplasm accession number of Tianlong 1 and why was it used instead of others?

Line 407 How was the Hoagland solution made or where did it come from?

Line 420 Where did the plasmid pCAMBIA1305-GFP come from.  A diagram of the plasmid and the NAC035 gene inserted into it is needed.  Need to explain what promoter was used where the GFP sequence is.  The transcriptional stop site should be indicated and what was its source.

Line 421-422 How was pCAMBIA1305-GFP transformed into E. coli DH5-alpha?

Line 422 How was pCAMBIA1305-GFP introduced into A. tumefaciens? Need a diagram of the construct

Line 423 How was the A. tumefaciens construct inserted into tobacco leaves?

Line 424 Where did you get the OsD53-mCherry construct?  If it came from one of the author of Ref: 58 just say so.

Line 425-426 What excitation frequencies were used for the different reporters?

Line 429 How did you get the full-length CDS sequence of GmNAC035?  It is not in the text.

Lines 429-430 How did you recombine the CDS sequence with pGBKT7? Cite the procedure or describe it.  A diagram of this construct is needed indicating the plasmid sequence, promoters used, the selectable marker used,  the GmNAC035 insert, what polyA signal was used.

Line 437-438 where did you get the pCAMBIA 1300 vector.  We need a diagram of it indicating the functional parts of it.

Line 439-440 What procedure was used to transfect A. tumefaciens and cite it.

Line 443 What procedure did you used to run the RT-qPCR and using what instrument

Line 453 briefly describe the MDA and SOD assay procedure or cite the procedure used.

Line 458 What was used to make up the NBT and DAB solutions? Water? Buffer? Cite the procedure.

Author Response

Response to Reviewer 2:

We acknowledge the reviewer2’s helpful comments and recommendations, which are valuable for improving our manuscript.

  1. Lines 89-91 are confusing. Could you re-write them to make the point easier to understand.

Reply: Thank you for your careful comments. We have redescribed this part of the content to make it easier to read and understand (New Line 98-100).

  1. Line 163 Why was pGBKT7 chosen? A diagram of the plasmid and insertion would make the text easier to understand. What procedure was used? Cite the procedure. What procedure was used to clone it into the yeast strain AH109?

Reply: Thank you for your comments. The pGBKT7 vector featured by a GAL4 DNA-binding domain (BD) and TRP1 auxotrophic marker, both enable fusion protein expression while minimizing autoactivation and grows on SD/-Trp medium. Besides, numerous studies have successfully used pGBKT7 to test autoactivation, including work on GmNAC181 (doi:10.1111/nph.18343). We have made detailed supplements to the materials and methods (New Line 468-480). The information regarding the pGBKT7 vector is available at Clontech.

(https://www.takarabio.com/documents/Vector%20Documents/pGBKT7%20Vector%20Information.pdf)

  1. Line 168 How was GmNAC35-GFP made? Cite the paper. A diagram of the construct would help understanding it. What was used as promoter? What was used as a stop signal? What was the MCS?

Reply: Thank you for your comments. We have added the vector structure in the new Figure S3. We used the pCAMBIA1305-GFP plasmid with the 35S promoter and the NOS Terminator.

  1. Line 168-170 How was the GmNAC035-GFP construct transformed into tobacco cells?  Cite the procedure. Where did the mCherry used in Fig. 3 E come from?

Reply: Thank you for your comments. We have improved this part of the content in the materials and methods section (New Line 450-467). OsD53-mCherry was used as a nuclear marker and came from Nanjing Agricultural University.

  1. Line 193 What is MS medium? If you made it from scratch you have to describe it.  If it was a commercial product then cite the product name. You need to describe how the salt was added to the medium. Was it added during the preparation or after the preparation (flooded onto the medium).

Reply: Thank you for your comments. We have improved this part of the content in the materials and methods section (New Line 494-500). The medium was prepared by mixing 2.22 g/L of PhytoTech M519 Murashige & Skoog Basal Medium with Vitamins (PhytoTech, Kansas, USA), 5 g/L of Gelzan™ CM (Sigma, Saint Louis, USA), and 10 g/L of sucrose (Sigma, Saint Louis, USA). After the mixture was prepared, the pH value was adjusted to 5.7-5.8 using 1 M KOH. The seedlings were then transferred to 1/2 MS medium mixed with different concentrations of NaCl (0 mM, 75 mM, 100 mM, and 125 mM) for salt stress treatment for 6 days.

  1. Line 220 Why use NBT and DAB?  What do they do? Which is a cell viability stain and which is an indicator of lipid peroxidation.

Reply: Hydrogen peroxide (H₂O₂) and superoxide anions (O₂⁻) are the main components of ROS in plants. 3, 3’-diaminobenzidine (DAB) and nitroblue tetrazolium (NBT) can be used to detect H₂O₂ and O₂⁻ respectively. DAB reacts with H₂O₂ to form a brown precipitate, while NBT reacts with O₂⁻ to produce a blue formazan precipitate (New Line 233-237). Using DAB and NBT to stain plant tissues allows for observing the aggregation sites of H2O2 and O2⁻, which can directly reflect the degree of oxidative damage to the plant.

  1. Fig5 panels C, D and E. Bar charts are a summary of the data. There should be a table in supplementary data with that data.

Reply: Thank you for your valuable comment. As you suggested, we have added the relevant data as New Table S5.

  1. Fig 6. Bar chart data should be in the supplementary files

Reply: Thank you for your valuable comment. As you suggested, we have added the relevant data as New Table S6.

  1. Line 393 what HMMsearch tool options were used if any

Reply: Thank you very much for your comments. We used HMMER (verson 3.1) with default parameters.

  1. Line 405 what was the germplasm accession number of Tianlong 1 and why was it used instead of others?

Reply: Thank you for your comments. Tianlong 1 is a high-yielding and stable-producing soybean with excellent quality and good disease resistance which is from Oil Crops Research Institute, Chinese Academy of Agricultural Sciences. Using Tianlong 1 as the material for soybean genetic transformation features a relatively high regeneration rate and transformation efficiency. Research related to GmNAC035 in soybeans is currently in progress.

  1. Line 407 How was the Hoagland solution made or where did it come from?

Reply: Thank you for your comments. The 1/2 Hoagland nutrient solution was purchased from Macklin Company. We have made supplements in the Materials and Methods section (New Line 434).

  1. Line 420 Where did the plasmid pCAMBIA1305-GFP come from.  A diagram of the plasmid and the NAC035 gene inserted into it is needed.  Need to explain what promoter was used where the GFP sequence is.  The transcriptional stop site should be indicated and what was its source.

Reply: Thank you for your comments. We have added the vector structure in the new Figure S3. We have also improved this part of the content in the materials and methods section (New Line 450-467).

  1. Line 421-422 How was pCAMBIA1305-GFP transformed into E. coli DH5-alpha?

Reply: Thank you for your comments. We have improved this part of the content in the materials and methods section (New Line 450-467).

  1. Line 422 How was pCAMBIA1305-GFP introduced into A. tumefaciens? Need a diagram of the construct

Reply: Thank you for your comments. We have improved this part of the content in the materials and methods section (New Line 450-467).

  1. Line 423 How was the A. tumefaciens construct inserted into tobacco leaves?

Reply: Thank you for your comments. We have improved this part of the content in the materials and methods section (New Line 450-467).

  1. Line 424 Where did you get the OsD53-mCherry construct?  If it came from one of the author of Ref: 58 just say so.

Reply: OsD53-mCherry was used as a nuclear marker and came from Nanjing Agricultural University (New Ref: 63).

  1. Line 425-426 What excitation frequencies were used for the different reporters?

Reply: Thank you for your comments. For GFP signals, the excitation wavelength was set at 488 nm, and the emission wavelength range was 505-530 nm. For mCherry signals, the excitation wavelength was 587 nm, and the emission wavelength was recorded in the range of 600-630 nm (New Line 465-467).

  1. Line 429 How did you get the full-length CDS sequence of GmNAC035?  It is not in the text.

Reply: Thank you for your comments. The full-length CDS sequence of GmNAC035 was amplified from Tianlong No.1 using the KOD FX (TOYOBO, Osaka, Japan), following the manufacturer's instructions (New Line 451-452).

  1. Lines 429-430 How did you recombine the CDS sequence with pGBKT7? Cite the procedure or describe it. A diagram of this construct is needed indicating the plasmid sequence, promoters used, the selectable marker used,  the GmNAC035 insert, what polyA signal was used.

Reply: Thank you for your comments. We have made detailed supplements to the materials and methods (New Line 468-480). The information regarding the pGBKT7 vector is available at Clontech.

(https://www.takarabio.com/documents/Vector%20Documents/pGBKT7%20Vector%20Information.pdf)

  1. Line 437-438 where did you get the pCAMBIA 1300 vector.  We need a diagram of it indicating the functional parts of it.

Reply: Thank you for your comments. The pCAMBIA 1300 vector is a laboratory preserved plasmid. We have added the vector structure in the new Figure S3.

  1. Line 439-440 What procedure was used to transfect A. tumefaciens and cite it.

Reply: Thank you for your comments. We have cited relevant references (New Ref: 61).

  1. Line 443 What procedure did you used to run the RT-qPCR and using what instrument

Reply: Thank you for your comments. We have improved this part of the content in the materials and methods section (New Line 440-444). According to the manufacturer’s protocol, TB Green Premix Ex TaqTM II (Takara, Tokyo, Japan) was used for RT-qPCR analysis of the cDNA, which was detected via the fully automatic medical PCR analysis system (Gentier 96R, TIANLONG, Shaanxi, China).

  1. Line 453 briefly describe the MDA and SOD assay procedure or cite the procedure used.

Reply: Thank you for your comments. The assays were performed using the Malondialdehyde (MDA) Content Assay Kit (Beijing Solarbio Science & Technology, Beijing, China), Peroxidase (POD) Activity Assay Kit (Beijing Solarbio Science & Technology, Beijing, China), and Superoxide Dismutase (SOD) Activity Assay Kit (Beijing Solarbio Science & Technology, Beijing, China), following the manufacturer’s protocol.

24.Line 458 What was used to make up the NBT and DAB solutions? Water? Buffer? Cite the procedure.

Reply: Thank you for your comments. Leaves were immersed in 5 mg/mL DAB (aqueous solution, pH 3.8) and 0.5 mg/mL NBT (aqueous solution, pH 7.5) in the dark for 20 hours.

Reviewer 3 Report

Comments and Suggestions for Authors

The manuscript “NAC Transcription Factor GmNAC035 exerts a positive regulatory role in enhancing salt stress tolerance in plants” by Shi et al. describes the identification of NAC TFs with potential roles in salt stress responses in soybean, and further characterization of a specific NAC TF (GmNAC035). The paper tells an interesting story, starting with genome-wide re-analysis of NAC TF genes in soybean, and mining of RNA-seq data to identify salt-stress responsive NAC TFs, followed by qRT-PCR to validate the expression results in response to salt stress. The research then continues with qRT-PCR to assess responsiveness of selected NAC TFs to two plant hormones. Taken together, this approach identified GmNAC035 as very responsive to salt and hormone treatment. Expression in yeast confirmed a likely role of NAC in transcriptional activation, and GmNAC035-GFP fusion protein expression in transgenic tobacco revealed localization in the nucleus, which is the expected location of a transcription factor.

The researchers then overexpressed GmNAC035 in Arabidopsis, which increased its salt tolerance. Next, the research revealed the possible causes for this increased salt tolerance:

  • elimination of ROS was demonstrated by staining and activity assays.
  • In addition, qRT-PCR of selected genes showed that overexpression of GmNAC035 increased expression of genes related to abiotic stress resistance.

Throughout this paper, I was impressed by the clarity of presentation, the logical flow, and the large amount of work that researchers have put in this manuscript. I’m reviewing papers fairly regularly, and I thought this paper stands out.

Suggestions

1) For qRT-PCR, I wish authors had use three biological replications, because technical replications will not show the variability between plants (though I understand that it would have been much more work). While the technical replications are mentioned in the Materials and Methods section, it would be good to also mention this in the figure legend of qRT-PCR figures, and any other figures that include replications; I suggest to always state how many and what type of replications were used.

2) The Materials and Methods would benefit from a little bit more detail throughout. For example,

“The CDS sequence of the GmNAC035 gene, excluding the stop codon, was recombined with the linearized pCAMBIA1305-GFP plasmid [56,57].” What kind of recombinant cloning (Gibson, in-fusion, …)

DAB and NBT: in what buffer? Are you concerned that the extreme pH of 3.5 (for DAB) could stress plants and change gene expression?

3) Very Minor points

“The plant-specific transcription factor NAC plays a crucial role in plant adaptation to abiotic stress conditions.”

How about:  Members of the plant-specific transcription factor family NAC play crucial roles in plant adaptation to abiotic stress conditions.

Introduction

“Similarly, in rice (Oryza sativa L.), the NAC genes OsNAC2, OsNAC5, OsNAC6 has been demonstrated…”

“Similarly, in rice (Oryza sativa L.), the NAC genes OsNAC2, OsNAC5, OsNAC6 have been demonstrated…”

Author Response

Response to Reviewer 3:

We wish to thank reviewer 3 for his/her strong support and valuable comments on this work.

  1. For qRT-PCR, I wish authors had use three biological replications, because technical replications will not show the variability between plants (though I understand that it would have been much more work). While the technical replications are mentioned in the Materials and Methods section, it would be good to also mention this in the figure legend of qRT-PCR figures, and any other figures that include replications; I suggest to always state how many and what type of replications were used.

Reply: Thank you for your valuable comment. Sorry, we didn’t describe it clearly. We conducted two independent experiments, and each sample had three technical replicates. The data presented here are representative of two biological replicates. According to your suggestion, we have provided explanations in both the relevant figure legends and the Materials and Methods section.

  1. The Materials and Methods would benefit from a little bit more detail throughout. For example,

“The CDS sequence of the GmNAC035 gene, excluding the stop codon, was recombined with the linearized pCAMBIA1305-GFP plasmid [56,57].” What kind of recombinant cloning (Gibson, in-fusion, …)

Reply: Thank you for your comment. We have made detailed supplements to the Materials and Methods section.

DAB and NBT: in what buffer? Are you concerned that the extreme pH of 3.5 (for DAB) could stress plants and change gene expression?

Reply: Thank you for your comment. NBT and DAB are in an aqueous solution. By using DAB and NBT to stain plant tissues, the accumulation sites of H₂O₂ and O₂⁻ can be observed, which can directly reflect the degree of oxidative damage suffered by the plants. The stained leaves will not be used for gene expression analysis.

  1. Very Minor points

“The plant-specific transcription factor NAC plays a crucial role in plant adaptation to abiotic stress conditions.”

How about: Members of the plant-specific transcription factor family NAC play crucial roles in plant adaptation to abiotic stress conditions.

Introduction

“Similarly, in rice (Oryza sativa L.), the NAC genes OsNAC2, OsNAC5, OsNAC6 has been demonstrated…”

“Similarly, in rice (Oryza sativa L.), the NAC genes OsNAC2, OsNAC5, OsNAC6 have been demonstrated…”

Reply: Thank you for your valuable comment. We have made revisions according to your suggestions.

Round 2

Reviewer 1 Report

Comments and Suggestions for Authors

1 How to keep the salt concentration from changing? Because the salt concentration will increase over time. You not explain clear in this response, pleased added it.

2 i m not see the changes in conclusions. pleased revise it.

Author Response

Response to Reviewer 1:

We acknowledge the reviewer1’s helpful comments and recommendations, which are valuable for improving our manuscript.

  1. How to keep the salt concentration from changing? Because the salt concentration will increase over time. You not explain clear in this response, pleased added it.

Reply: Thank you for your comments. In this study, the salt stress treatment was conducted in strict accordance with established methodologies from previous research [1, 2]. Specifically, this salt stress treatment was carried out in a plant growth chamber with a humidity of approximately 60%, under a 16-hour dark/8-hour light photoperiod, rather than in an unpredictable conditions of an open-field environment. Moreover, we completed the sampling within 48 hours after the stress treatment. Therefore, the evaporation of the salt solution was extremely minimal, which effectively ensured that the salt concentration remained relatively stable during the experiment.

  1. i m not see the changes in conclusions. pleased revise it.

Reply: Thank you for your valuable feedback. I have carefully revised the conclusion section. In the updated version, I have streamlined the content to make it more concise and clear (New Conclusions).

References:

  1. Xu C, Shan J, Liu T, et al. CONSTANS-LIKE 1a positively regulates salt and drought tolerance in soybean. Plant Physiol. 2023;191(4):2427-2446.
  2. Wang C, Li X, Zhuang Y, et al. A novel miR160a-GmARF16-GmMYC2 module determines soybean salt tolerance and adaptation. New Phytol. 2024;241(5):2176-2192.
